# An Obstacle Detection Algorithm Suitable for Complex Traffic Environment

**Guantai Luo [1], Xinwei Chen [2], Wenwei Lin [1], Jie Dai [1], Peidong Liang [1] and Chentao Zhang [1,3,\*]**

[1] Fujian (Quanzhou)-HIT Research Institute of Engineering and Technology, Quanzhou 362000, China; lgt2091@hitqz.com (G.L.); lww2127@hitqz.com (W.L.); dj2169@hitqz.com (J.D.); lpd0004@hitqz.com (P.L.)
[2] Industrial Robot Application of Fujian University Engineering Research Center, Minjiang University, Fuzhou 350121, China; chen_xinwei@126.com
[3] Department of Instrumental and Electrical Engineering, Xiamen University, Xiamen 361102, China
[\*] Correspondence: zhangct@xmu.edu.cn; Tel.: +86-15959285107

**Abstract:** For the task of obstacle detection in a complex traffic environment, this paper proposes a road-free space extraction and obstacle detection method based on stereo vision. The proposed method combines the advantages of the V-disparity image and the Stixel method. Firstly, the depth information and the V-disparity image are calculated according to the disparity image. Then, the free space on the road surface is calculated through the RANSAC algorithm and dynamic programming (DP) algorithm. Furthermore, a new V-disparity image and a new U-disparity image are calculated by the disparity image after removing the road surface information. Finally, the height and width of the obstacles on the road are extracted from the new V-disparity image and U-disparity image, respectively. The detection of obstacles is realized by the height and width information of obstacles. In order to verify the method, we adopted the object detection benchmarks and road detection benchmarks of the KITTI dataset for verification. In terms of the accuracy performance indicators quality, detection rate, detection accuracy, and effectiveness, the method in this paper reaches 0.820, 0.863, 0.941, and 0.900, respectively, and the time consumption is only 5.145 milliseconds. Compared with other obstacle detection methods, the detection accuracy and real-time performance in this paper are better. The experimental results show that the method has good robustness and real-time performance for obstacle detection in a complex traffic environment.

**Keywords:** obstacle detection; V-disparity image; U-disparity; Stixel-World

## 1. Introduction

With the milestone leaps in computer computing power and the continuous improvement of sensor precision, more and more researchers have turned their attention to Advanced Driving Assistance Systems (ADASs) [1]. In the process of implementing ADASs, real-time and accurate obstacle detection algorithms are undoubtedly the top priority. The obstacle detection algorithm detects the position of obstacles on the road which provides important information for the automatic driving decision-making system [2]. A large number of algorithms based on various sensors (such as cameras, lidar and millimeter-wave radar) have been proposed to detect roads and obstacles on the road [3]. Lidar can convert the surrounding environment information into point cloud data to detect obstacles around the road and even make a preliminary estimate of the shape of the obstacle. However, for small obstacles in traffic scenes, lidar risks making missing or false detections. Furthermore, its high price also limits its application range. Millimeter-wave radar is a better choice for multi-scale and multi-target detection, its widespread use is complicated due to its short detection distance, being easily affected by environmental changes, and being expensive [4].

Compared to other sensors, vision sensors provide intuitive image data and there is no signal interference between vision sensors. Therefore, vision sensors are widely deployed in ADASs [5]. In addition, the real-time processing of image matching has been realized

due to the improvement of GPU parallel computing technology. The research of obstacle detection methods based on vision has gradually become a research field that has attracted lots of attention.

At present, there are two popular research methods for obstacle detection based on binocular vision. The two research directions are projecting the disparity image in the image coordinate system and the polar coordinate system, respectively. The disparity image projection method based on the image coordinate system usually converts the disparity image into a depth image. The disparity image is, respectively, projected in the horizontal and vertical directions to calculate the V-disparity image and the U-disparity image [6]. Then, road information is extracted from the V-disparity image by the Hough transform or line fitting algorithm. The V-disparity image and the U-disparity image are recalculated after removing the road information in the disparity image. Finally, the obstacle height and width are calculated from the V-disparity image and the U-disparity image, respectively. The object detection area is projected onto the original image to realize obstacle detection. The obstacle detection method based on the U-disparity image and V-disparity image has a good effect on plane obstacle detection. However, this method is computationally expensive and has poor anti-interference performance.

In contrast to the disparity image projection method based on the image coordinate system, the polar coordinate-based disparity image projection method generates a polar occupancy grid (POG) by projecting the disparity image in polar coordinates [7]. Then, the background is subtracted from the POG by setting a threshold. Furthermore, the boundary of road-free space and the lower boundary of obstacles are calculated using the DP algorithm. The optimal dividing line for the difference between the foreground (target) and background is calculated by using DP again. The upper boundary of the obstacle is extracted through the optimal dividing line. Finally, the upper and lower boundaries of all obstacles are combined to generate Stixel-World to realize the detection of an obstacle. Although the Stixel-World method has good performance in obstacle detection, this method needs to run the DP algorithm twice, which makes its real-time performance slightly insufficient. Furthermore, the obstacle detection accuracy of this method depends on the result of the road shape fitting.

In order to effectively detect obstacles on the road, Labayrade et al. proposed the concept of a V-disparity image for obstacle detection on a non-flat road [8]. Hu et al. proposed the concept of the U-disparity image to make up for the shortcomings of the algorithm proposed by Labayrade [6]. Zhang et al. combined the straight-line fitting algorithm and the Hough detection algorithm to process the V-disparity image, which improved the original algorithm [9]. The road surface and the obstacle area can be projected into line segments using the U-disparity image and V-disparity image, which transforms the plane detection into line segment detection. This method can significantly reduce the computational complexity and improve the accuracy of detection. Badino et al. proposed the Stixel-World algorithm for the flexible representation of three-dimensional traffic scenes, which effectively improved the detection speed of the obstacle [10]. Combined with the SGM algorithm proposed by Hirschmüller [11], the algorithms of these two directions have greatly improved the accuracy and real-time performance of obstacle detection. However, for actual complex traffic scenarios such as crowded vehicles, cluttered backgrounds, and mottled shadows, the accuracy and real-time performance of the above detection methods still fall short of fully meeting the actual requirements of applications.

In order to improve the real-time performance and accuracy of algorithms, we propose an obstacle detection algorithm with higher real-time performance and accuracy of obstacle detection. In this paper, we propose an improved obstacle detection algorithm based on Stixel-World combined with the U-disparity and V-disparity images. In contrast to existing Stixel series algorithm for the estimation method of obstacle height information, we developed an obstacle height estimation method based on the U-disparity and V-disparity images, and improved the estimation speed of the free space on the road. The depth information and the V-disparity image are calculated according to the disparity

image. The free space on the road is calculated through the RANSAC algorithm and DP algorithm. Furthermore, a new V-disparity image and a new U-disparity image are calculated by the disparity image after removing the road surface information. Finally, the height and width of obstacles on the road are extracted from the new V-disparity image and U-disparity image, respectively. The detection of obstacles is realized by the height and width information of obstacles. Experiments show that this method can be used under the influence of crowded vehicles, cluttered backgrounds, mottled shadows, and bad weather in actual traffic scenes.

The rest of the paper is arranged as follows: the second section introduces the proposed method in detail; the third section uses complex traffic scenes in the KITTI dataset to verify the experimental results of the proposed method; finally, the fourth section summarizes the paper.

## 2. Road-Free Space Extraction and Obstacle Detection

The method we propose includes the two following parts: the calculation of free space and the estimation of obstacle height information. The depth information and the V-disparity image are calculated according to the disparity image. The free space on the road surface is then calculated through RANSAC algorithm and DP algorithm. Furthermore, a new V-disparity image and a new U-disparity image are calculated by the disparity image after removing the road surface information. Finally, the height and width of obstacles on the road are extracted from the new V-disparity image and U-disparity image, respectively.

### 2.1. Stereo Geometry

Figure 1 shows a stereo vision system model. A point $p$ $(X, Y, Z)$ is projected into the left and right views to obtain the image coordinates $(u, v)$. In order to simplify this issue, we assume that $\theta$ is the pitch angle of the camera and $h$ is the height of the camera above the ground. $b$ is the baseline distance between the left and right cameras. $f$ is the focal length of the left and right cameras in pixels. $K$ represents the intrinsic parameter matrix of the camera and its expression is shown in Equation (1).

$$K = \begin{pmatrix} f_x & 0 & c_x \\ 0 & f_y & c_y \\ 0 & 0 & 1 \end{pmatrix} \Rightarrow \begin{cases} u_0 = c_x \\ v_0 = c_y \\ f = f_x = f_y \end{cases} \tag{1}$$

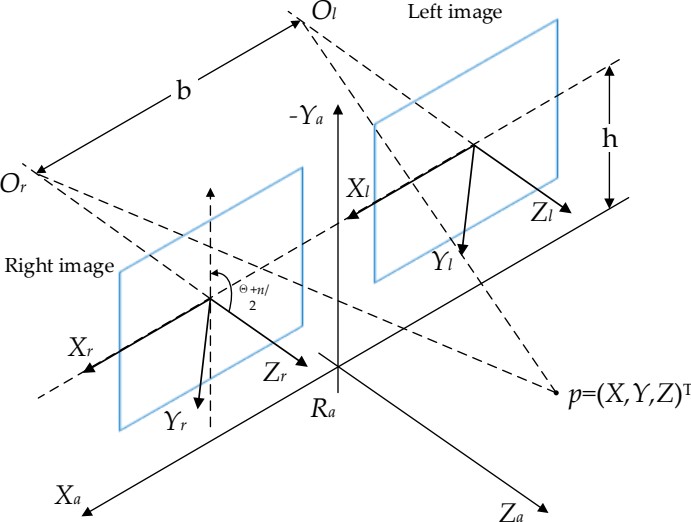

**Figure 1.** Stereo vision system.

According to the above parameters, the image coordinates (*u*, *v*) can be directly calculated by Equation (2).

$$
\begin{aligned}
u_{l,r} &= u_0 + f_x \frac{X \pm b/2}{(Y+h)\cdot\sin\theta + Z\cdot\cos\theta} \\
v &= v_0 + f_y \frac{(Y+h)\cdot\cos\theta - Z\cdot\sin\theta}{(Y+h)\cdot\sin\theta + Z\cdot\cos\theta}
\end{aligned}
\tag{2}
$$

Then, the disparity image coordinate (*U*, *V*) and the disparity value Δ can be calculated by Equation (3).

$$
\begin{cases}
U_{l,r} = u_{l,r} - u_0 = f\dfrac{X \pm b/2}{(Y+h)\cdot\sin\theta + Z\cdot\cos\theta} \\[2mm]
V = v - v_0 = f\dfrac{(Y+h)\cdot\cos\theta - Z\cdot\sin\theta}{(Y+h)\cdot\sin\theta + Z\cdot\cos\theta} \\[2mm]
\Delta = u_l - u_r = \dfrac{fb}{(Y+h)\cdot\sin\theta + Z\cdot\cos\theta}
\end{cases}
\tag{3}
$$

Similarly, according to Equation (3), the depth information can be calculated by:

$$
\Delta = \frac{fb}{Z} \quad D = Z = \frac{fb}{\Delta}
\tag{4}
$$

As shown in Figure 2, the disparity image (b) is calculated based on the left image (a) through Equation (4). Image (c) is a pseudo-color image of depth image.

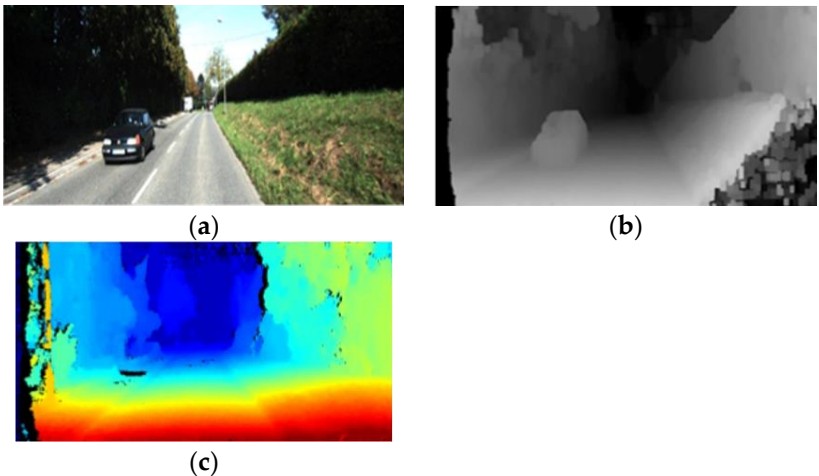

(a)

(b)

(c)

**Figure 2.** Disparity image calculated based on stereo vision system: (**a**) left image; (**b**) disparity image; and (**c**) pseudo-color image of depth image.

### 2.2. Extraction of Free Space

As shown in Figure 3, the disparity image is projected to the horizontal and vertical directions on the image coordinate system (*U*, *V*) to obtain the V-disparity image and the U-disparity image. For the V-disparity image, the value in the V-disparity image represents the cumulative sum of the same disparity value in the same row in the disparity image, which converts the horizontal plane in the original view into line segments (obstacles are usually in a vertical position and the road surface occupies a larger part [12]. Therefore, the projection of the road surface in the V-disparity image is an obvious oblique line segment). The value of the U-disparity image represents the cumulative sum of the disparity image and has the same disparity value in the same column, and the vertical plane of the original image is converted into line segments.

When the road surface is transformed into oblique line segments, the Hough transform, least square method [13] or RANSAC algorithm [14] are used to extract straight line segments in the V-disparity image to obtain road surface information. Since real traffic scenarios usually contain noise and roads are usually not flat surfaces, we therefore use the RANSAC algorithm to extract the straight line projected onto the road, which can reduce the interference of noise and improve the accuracy and real-time detection.

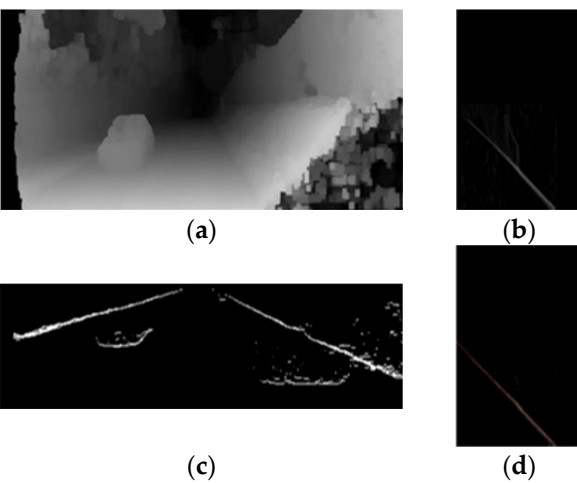

(a)　　　　　　　　　　　　　(b)

(c)　　　　　　　　　　　　　(d)

**Figure 3.** Projection result of the disparity image in the horizontal and vertical directions: (**a**) disparity image; (**b**) V-disparity image; (**c**) U-disparity; and (**d**) line of road surface.

After extracting the road fitting straight line, we used the method proposed by Pfeiffer D. et al. [15] to directly use the disparity image to obtain the score image. In the score image, the DP algorithm was used to extract free space. Figure 4 shows the score image. Then, we calculated the total cost function of each pixel in each column of the disparity image, and the score of roads and obstacles can be calculated by:

$$
\begin{aligned}
\Gamma_v &= \alpha_1 \Gamma_v^R + \alpha_2 \Gamma_v^O, \text{ with} \\
\Gamma_v^O &= \sum_{v=v_b-h_v}^{v_b} |d_R(v_b) - d_v| \\
\Gamma_v^R &= \sum_{v=v_b}^{V} |d_R(v) - d_v|
\end{aligned}
\tag{5}
$$

where $\Gamma_v$ represents the score of pixel point $d_v$ on the disparity image; $\Gamma_v^O$ represents obstacle evidence; $\Gamma_v^R$ represents road evidence; $d_R(v_b)$ represents base point disparity; and $d_R(v)$ represents given disparity profile.

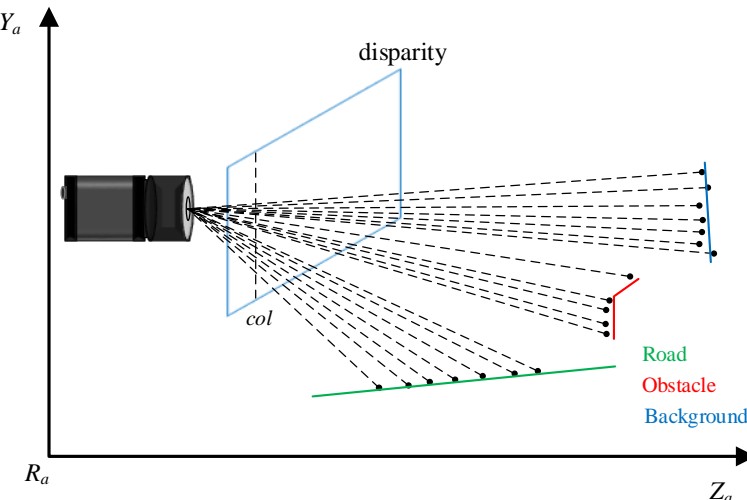

**Figure 4.** Score image.

The final score is equal to the sum of roads and obstacles. A column in the disparity image includes the points of road, obstacle and background. The small score means that the object may be road or obstacle, and the current point is the junction of the two. All the

junction points are extracted by the DP algorithm to obtain the best free space dividing line and extract the free space on the road.

### 2.3. Height Information

We use the V-disparity image with the road surface information removed to calculate the height information of obstacles. According to the aforementioned method, the calculation formula of the obstacle height information is shown in Equation (6).

$$Z = \frac{bf}{\Delta} \quad Y = \frac{(v - v_0)Z}{f} - h \tag{6}$$

where $Y$ and $Z$ represent the height and depth information of the target obstacle in the disparity image, which can be calculated by the $\Delta$ and $v$ parameters of the V-disparity image. Finally, setting a threshold for the obstacle height to remove the error data and projecting the obstacle height into the disparity image to calculate the upper boundary of the obstacle.

### 2.4. Obstacle Detection

After calculating the free space on the road, the lower boundary of the obstacle can be obtained. Considering the fact that the same obstacle in the traffic scene has similar disparity values in the corresponding area in the disparity image, the free space results on the road surface can be filtered by setting the threshold value of the disparity value. The filtered disparity value takes the obstacle in the V-disparity image as the base point and further filters the obstacle line segment to realize the obstacle height estimation. Similarly, the disparity value of the U-disparity image is filtered to estimate the width of the obstacle. As shown in Figure 5, it is the road-based obstacle detection result.

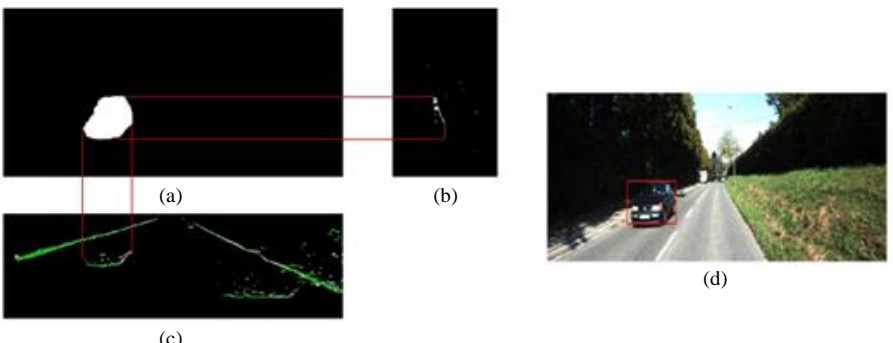

**Figure 5.** Obstacle detection result: (**a**) obstacle disparity; (**b**) vertical line segment fitting; (**c**) horizontal line segment fitting; and (**d**) obstacle detection.

## 3. Experiment

### 3.1. Datasets and Evaluation Index

The KITTI dataset [16] is an authoritative dataset used in computer vision for mobile robotics and autonomous driving. The KITTI dataset has lots of images in different scenarios which provide the verification and test benchmarks for testing various algorithms. In order to evaluate the effectiveness of our proposed method, we used the road detection benchmarks in the KITTI autonomous driving dataset for verification in our experiments. The resolution of the binocular image of the detection benchmark is 1242 × 375, and it mainly contains 1500 images of three typical traffic scenes including 600 images in the urban scene, 400 images in the highway, and 500 images in the second-class road. The KITTI dataset consists of three parts: the training set, test set and validation set. In order to better test our proposed algorithm, we filter out three different non-overlapping parts from the dataset: training set (300), validation set (100), and test set (300), and the proportions of the three typical traffic scenarios are the same.

In this paper, according to the actual ground and detection results, we mark each pixel in the detection image as one of the four conditions of TN, FN, FP, and TP. Then, we use four pixel-wise measures which including the quality, detection rate, detection accuracy and effectiveness to test the effectiveness of the proposed method [17]. The specific definitions are shown in Tables 1 and 2. In addition, in order to reduce error caused by the ground truth generated by manual labeling, we discarded boundary pixels in the evaluation process.

**Table 1.** Contingency table.

|                    | **Road** | **Non-Road** |
| ------------------ | -------- | ------------ |
| Detected road      | TN       | FN           |
| Detected non-road  | FP       | TP           |

**Table 2.** Definition of Measures.

| **Pixel-Wise Metric** | **Definition** |
| --------------------- | -------------- |
| Quality               | $Q = \frac{TP}{TP+FP+FN}$ |
| Detection rate        | $DR = \frac{TP}{TP+FP}$ |
| Detection accuracy    | $DA = \frac{TP}{TP+FN}$ |
| Effectiveness         | $E = \frac{2DR \cdot DA}{DR+DA}$ |

### 3.2. Calculate the Free Space

Section 3 introduces the main steps of the proposed algorithm to extract free space on the road. We chose the work in [8,15] as the baseline algorithm. The first baseline algorithm [8] was chosen because it is a representative work of the flat road assumption, which is the most widely used model to date. In addition, to the best of our knowledge, the work in [8] has the lowest computational complexity among all the reported three-dimensional road detection methods. The second baseline algorithm [15] is a typical work in the research direction of Stixel, and subsequent research in the Stixel algorithm is derived from this work.

The work in [8] expresses the vertical road contour formula as the inclined straight line of the flat road and the piecewise linear curve of the non-flat road in the V-disparity image. Since the KITTI dataset contains flat road scenes, we use the flat road assumption for the baseline algorithm to test. The main problem in work [8] is that it heavily relies on data fitting and does not deal with lots of input data noise. Both roads and obstacles will be projected as straight lines in the V-disparity image. Therefore, a line extraction technique such as the Hough transform will eventually extract multiple lines, which makes it difficult to identify the line (free-space) or line cluster (non-free-space) corresponding to the actual road contour. Our proposed algorithm uses the RANSAC algorithm for straight line fitting to extract the line of the road surface in the V-disparity image. The method in this paper reduces both the computational complexity and the influence of noise on road surface extraction.

The Stixel algorithm has higher accuracy for free space calculation and obstacle detection. The algorithm uses two DP algorithms to calculate the obstacle base point and estimate the obstacle height, respectively, which reduces the real-time performance. Therefore, this paper uses the RANSAC algorithm to fit and extract the line segments represented by the free space in the V disparity image. Then, the method in this paper removes the parallax of non-obstacles on the ground, which reduces the detection range of the ground in the original parallax image and improves the real-time performance of the algorithm. Finally, the score image is used to distinguish the obstacles and the ground. The DP algorithm is used to traverse the entire image to extract the dividing line. The extraction result is shown in Figure 6a. The template corresponding to the free space is mapped to the original left-view image, shown in Figure 6b.

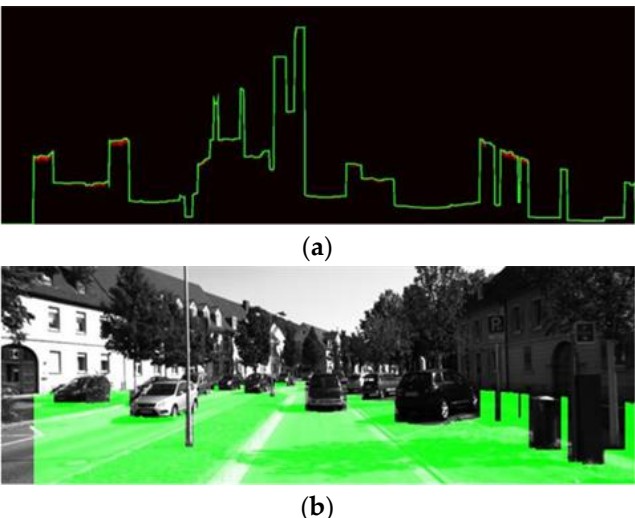
(**a**)

(**b**)

**Figure 6.** Result of extracted free space: (**a**) dividing line of free space and obstacle; and (**b**) result of free space on ground.

### 3.3. Estimate Height Information

We used the Stixel algorithm for height estimation, which followed the same process as the calculation of free space. Firstly, the boundary between the foreground and the background is extracted. The contour line of the height information is obtained by traversing the image with the DP algorithm. The Stixel-World algorithm is effective for obstacle detection. However, it cannot distinguish obstacles near the ground with similar disparity and the calculation is complicated. The V-disparity image is used to extract the height information of the obstacles in the region of interest. The previous binocular system model shows that the corresponding obstacle in the free space is expressed as a line segment that has an intersection with the ground contour line on the line of the V-disparity image. The effective extraction of this line segment can accurately obtain the height information corresponding to the obstacle. However, it is difficult to make the entire area of the same obstacle in the same vertical plane with the same distance from the camera in the real traffic scenario. This problem will cause the projection of the same obstacle in the V-disparity image and U-disparity image to be multiple discontinuous straight-line segments. Additionally, it is more difficult and computationally complex to completely extract all the pixels of the obstacle in the UV-disparity image. To solve this problem, we used a region growing algorithm to fit the pixels of the disparity image corresponding to the obstacle in the UV disparity image, which effectively improves the accuracy of obstacle detection in free space. The result is shown in Figure 7.

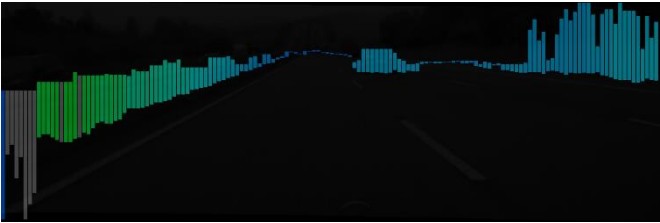

**Figure 7.** Result of height segmentation.

### 3.4. Results

In order to fully test the effectiveness of the proposed algorithm, the proposed algorithm was compared with two existing obstacle detection algorithms. According to 4 parameters in Table 2, 3 algorithms are run with the same test dataset. In addition to these 4 parameters, the average calculation time of the free space was also taken into account with the same test dataset. All experiments in this paper were run on Intel's Core

i7-9750H CPU, based on Microsoft's Visual Studio 2017 and open source programming library OpenCV4.4.0 compilation environment. The final results are shown in Table 3. It can be seen from Table 3 that the method in this paper is superior to the Stixel-origin algorithm and the V-disparity based method in terms of quality, detection rate, detection accuracy, effectiveness and speed. This proves that the method proposed in this paper has good robustness, effectiveness and accuracy.

**Table 3.** Result of different algorithm.

| Method | Q | DR | DA | E | T (ms) |
|---|---|---|---|---|---|
| Stixel-origin | 0.792 | 0.849 | 0.923 | 0.884 | 5.673 |
| V-disparity-based method | 0.781 | 0.921 | 0.831 | 0.874 | 6.345 |
| Our method | 0.820 | 0.863 | 0.941 | 0.900 | 5.145 |

As shown in Figure 8, it is the processing process and detection results of the method in this paper for free space extraction and obstacle detection. It can be seen from the figure that the method in this paper effectively detects the road and obstacles on the road, which provides key information for the automatic driving system.

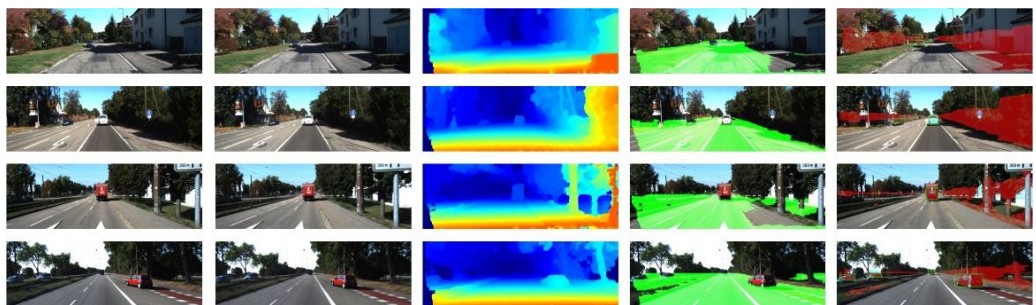

**Figure 8.** More about the detection results of the KITTI dataset in complex traffic scenarios. From left to right are the right image, left image, disparity image, free space extraction, and obstacle detection.

## 4. Conclusions

In this paper, an obstacle detection and estimation algorithm based on Stixel vision was proposed. Compared with the work in [8], our method computationally reduces the complexity of making an effective Hough space and improves the accuracy of road surface extraction results. Compared with the Stixel-World algorithm, the method in this paper improves the real-time performance of obstacle height information estimation in complex traffic scenes. The experimental results show that the proposed method achieves both robustness and real-time obstacle detection. The method in this paper can realize the real-time detection of obstacles and roads in the process of vehicle driving, which is helpful for automatic obstacle avoidance and early warning. The method in this paper provides new detection ideas for the field of autonomous driving, which will help the development of the field of autonomous driving.

At present, this method has only been verified on public datasets. In the future, we will to build experimental vehicle equipment and collect data for further analysis. We will try to use more than two cameras for obstacle detection or use auxiliary devices such as ultrasonic wave distance-measuring sensors to improve the performance of the detection algorithm. In addition, this paper only conducts research on road and obstacle detection and does not classify obstacles. On the basis of our research results, we will carry out related research work on obstacle classification based on deep learning. We believe this research direction will contribute to the development of the field of autonomous driving.

**Author Contributions:** Conceptualization, W.L. and G.L.; methodology, G.L. and C.Z.; software, P.L.; validation, X.C.; formal analysis, G.L., J.D. and C.Z.; investigation, J.D.; resources, X.C. and C.Z.; data

curation, W.L.; writing—original draft preparation, C.Z., X.C. and G.L.; writing—review and editing., G.L. and X.C.; visualization, P.L.; supervision, J.D. and C.Z. All authors have read and agreed to the published version of the manuscript.

**Funding:** This research was funded by the Industrial Robot Application of Fujian University Engineering Research Center, Minjiang University under Grant No. MJUKF-IRA1903, and the Scientific and Technological Program of Quanzhou City under Grant No. 2020FX02.

**Institutional Review Board Statement:** Not applicable.

**Informed Consent Statement:** Not applicable.

**Data Availability Statement:** The data presented in this study are available on request from the corresponding author. The data are not publicly available due to privacy.

**Conflicts of Interest:** The authors declare no conflict of interest.

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
