# Peer review of "An Obstacle Detection Algorithm Suitable for Complex Traffic Environment"

_wevj, doi:10.3390/wevj13040069_

Round 1
Reviewer 1 Report
This manuscript proposed an approach using V-disparity image and the Stixel method to detect the free space and obstacles on the road. The proposed approach was verified on the KITTI dataset. The experimental results suggest that this approach outperforms other commonly used approaches in terms of quality, accuracy, effectiveness and with high time efficiency. This topic is interesting, however, the following concerns prevent me from recommending the manuscript be published at the current stage.
- The language of the manuscript needs to improved and revised by some linguistic professionals;
- The descriptions of the obstacle detection in the introduction section seem a little bit length: What is the point of illustrating the two different approaches presented in Figure 1 and Figure 2? Which approach will be adopted by the manuscript?
- The authors mixed the road free space extraction and obstacle detection, are there any connections between the two approaches? If yes, please explain.
- What are the differences between the image (b) and image (c), I cannot find any differences between the two images.
- What is the contributions of the research in terms of automatic driving field?
Author Response
Dear reviewer:
we greatly appreciate that you have spent your valuable time reviewing our manuscript. In response to your suggestions for revisions, we have made revisions and explanations one by one. Attached is the details.

Reviewer 2 Report
The authors propose an improved obstacle detection algorithm based on Stixel World combined with U-disparity and V-disparity images. The authors developed an obstacle height estimation method based on the UV-disparity images, improving the estimation speed of the free space on the road. In overall, the paper structure is adequate, even though it needs some adjustments.
- Read the text carefully; there are some typo errors.
- Do not forget to use capital letters when referencing Equation.
- Use the same font size for equations. The font size of the equations is big compared to text font size.
- It seems that the text was changing from one format to another. For example, the authors use Roman numerals for referencing Table II, and for referencing Table 3, they use Arabic numerals.
- I am confused about this affirmation: "we filter out three different non-overlapping parts from the dataset: training set (300), cross-validation set (100), and test set (300)". The authors are confusing concepts, a validation set is different from a cross-validation experiment.
- In Table 1, the terms Free-space and non-free-space are confusing. Why did you not use the same terms of the referenced paper? Road/non-road
- In Table 2, the authors decided to use the variable P for the Detection Rate. Why do the authors not use the variable DR or the term Precision? The same for Detection Accuracy (Recall) and Effectiveness (F1 score).
- Include the ground truth in Figure 10.
- The paper's title is about obstacle detection, but there are no experiments about the detection of the different obstacles like vehicles, pedestrians, etc. The results show the performance on the detection of non-roads.
Author Response

(The authors gave the same response as above.)

Round 2
Reviewer 1 Report
This is the revised manuscript that I reviewed before. This version indicates a great improvement compared with the previous version. I laud authors hard work on revising the manuscript. From my point of view, the authors have addressed all the concerns that reviewers raised. I recommend that the manuscript should be accepted.
Reviewer 2 Report
I appreciate all the effort made to correct all things pointed out in the last review.